# Wind Preview-Based Model Predictive Control of Multi-Rotor UAVs Using LiDAR

**DOI:** 10.3390/s23073711

**Published:** 2023-04-03

**Authors:** Arthur P. Mendez, James F. Whidborne, Lejun Chen

**Affiliations:** 1Dynamics Simulation and Control Group, Cranfield University, Cranfield MK43 0AL, UK; 2Department of Electronic and Electrical Engineering, University College London, London WC1E 6BT, UK

**Keywords:** MPC, remote wind sensing, gust rejection, preview control, optimal control, quadrotor, VTOL aircraft

## Abstract

Autonomous outdoor operations of Unmanned Aerial Vehicles (UAVs), such as quadrotors, expose the aircraft to wind gusts causing a significant reduction in their position-holding performance. This vulnerability becomes more critical during the automated docking of these vehicles to outdoor charging stations. Utilising real-time wind preview information for the gust rejection control of UAVs has become more feasible due to the advancement of remote wind sensing technology such as LiDAR. This work proposes the use of a wind-preview-based Model Predictive Controller (MPC) to utilise remote wind measurements from a LiDAR for disturbance rejection. Here a ground-based LiDAR unit is used to predict the incoming wind disturbance at the takeoff and landing site of an autonomous quadrotor UAV. This preview information is then utilised by an MPC to provide the optimal compensation over the defined horizon. Simulations were conducted with LiDAR data gathered from field tests to verify the efficacy of the proposed system and to test the robustness of the wind-preview-based control. The results show a favourable improvement in the aircraft response to wind gusts with the addition of wind preview to the MPC; An 80% improvement in its position-holding performance combined with reduced rotational rates and peak rotational angles signifying a less aggressive approach to increased performance when compared with only feedback based MPC disturbance rejection. System robustness tests demonstrated a 1.75 s or 120% margin in the gust preview’s timing or strength respectively before adverse performance impact. The addition of wind-preview to an MPC has been shown to increase the gust rejection of UAVs over standard feedback-based MPC thus enabling their precision landing onto docking stations in the presence of wind gusts.

## 1. Introduction

Unmanned Aerial Vehicles (UAVs) are being increasingly employed for autonomous operations due to their cost-effectiveness in applications such as aerial inspection, 3-D mapping, search and rescue, delivery, and precision agriculture. Their small size and use in urban environments makes them susceptible to wind gust which can cause significant deviation from their intended flight path. This poses a safety risk to the UAV and its surroundings, particularly in applications such as aerial inspection and delivery. Furthermore, the autonomous nature of their operations entails the need for accurate position control to enable the UAVs to successfully dock onto a charging station. Gust resilience is therefore a necessary requirement for the safe and reliable operations of autonomous UAVs. A powerful method to impart gust resilience to UAVs is through the development of flight control systems that can sense and compensate the disturbance by making real-time adjustments to the UAV’s attitude and trajectory to ensure that it remains stable and on course.

Most research on gust rejection of UAVs tackles the problem of wind disturbance by first utilising robust model-based filters or observers to estimate the disturbance acting on the aircraft and then using a suitable feedback controller to provide the necessary compensation. Ref. [1] develops a generalised extended state observer to estimate the wind disturbance and utilise an attitude tracking backstepping controller for disturbance rejection. Refs. [2,3] model the effects of wind on the UAV’s dynamics to explicitly estimate the wind velocity acting on the aircraft for disturbance rejection. Refs. [4,5,6,7] propose the use of active disturbance rejection control, where extended state observers are utilised to estimate the total disturbance acting on the aircraft followed by the adjustment of an adaptive feedback controller to reject the effects of the disturbance. Due to the utilisation of observers in all these methods, the wind disturbance needs to propagate throughout the entire system before it is sensed and compensated for. This inevitably adds a delay to the compensation which decreases the potential performance improvement. This becomes more significant for larger UAVs due to their increased actuator lag and slower aircraft dynamics. An alternative to these sole feedback-based methods is preview-based disturbance rejection. Here, future wind disturbance information is used to compute the necessary control compensation either preemptively (to account for delays) or in real-time. A suitable methodology to utilise this preview information for optimal gust rejection is Model Predictive Control (MPC).

MPC was initially widely used in the petro-chemical industry due to its ability to easily handle dynamic constraint manipulation. This allowed the operators to react to rapid changes in the marketplace while keeping the plant at the most economically optimal operating condition [8]. With its ability to easily handle multivariable control applications, MPC started gaining popularity in flight control of UAVs due to the ready access of lighter, faster, and cheaper computers to run numeric algorithms for solving its dynamic optimization problem. Most implementations of MPC in UAVs focused on guidance control [9,10,11,12], where the MPC’s dynamic optimization and constraint handling allowed for optimal trajectory generation in dynamic and unknown environments. In Refs. [13,14], the MPC algorithm was leveraged to compute UAV trajectories which optimize both for system dynamics and perception objective that aims to maximize visibility of a point-of-interest to enable reliable vision-based tracking for navigation. Ref. [15] found improvements in trajectory tracking performance under wind disturbances through the use of a non-linear MPC over a linear MPC due to the inclusion of full system dynamics. Refs. [16,17,18,19] estimate the disturbances on the vehicle for offset-free tracking with the MPC. Ref. [18] uses an MPC in the position control loop of a tail-sitter Vertical-Takeoff-Landing (VTOL) UAV where wind is considered as a measurable disturbance and feedforward control is incorporated into the MPC to reject its effects. While MPC with measured disturbance feedforward can improve its gust rejection properties compared to purely feedback based methods, this can be taken a step further by using a preview of the incoming wind for its rejection. Ref. [20] developed an MPC strategy that estimates and stores the wind disturbance information of a leader quadrotor to be used as the disturbance preview for the follower quadrotor thus increasing its disturbance rejection. While this utilises the MPC’s future optimization to reject the incoming disturbance, the strategy is limited to situations where past information of steady wind disturbance fields is available and hence is not suitable for the rejection of dynamic wind gusts. In this work, the incoming wind disturbance on a quadrotor UAV is estimated using remote measurements from external ground-based sensors. The remote measurements are then used to predict the future disturbance profile on the aircraft. This preview information is then utilised by an MPC for optimal disturbance rejection over its prediction horizon. The feasibility of this preview-based strategy is mainly limited by the availability of accurate preview information, the possibility for which has increased due to ongoing development of remote wind sensing instruments such as LiDAR.

LiDAR (Light Detection and Ranging) can be broadly split into two types: ranging LiDAR s which are used for surveying/mapping, and atmospheric LiDAR s which can be used to measure wind speed. The latter utilises the Doppler frequency shift of back-scattered light from airborne particles to measure the wind speed. Here in this paper, the term LiDAR is used to exclusively refer to the wind measuring subgroup of LiDAR s. Wind energy was the first major industry to make use of the remote wind sensing capability of LiDAR  [21,22,23,24,25], the main idea being to utilise a forward-looking LiDAR mounted on the nacelle of the wind turbines to measure incoming wind disturbances. This preview information is then used by the turbine’s blade pitch controller to effectively mitigate against gust loads, decrease turbine fatigue, and increase its power output [23,24,25,26]. The measurement range of these remote sensors also facilitates the study and modelling of air flow in complex terrains and wind farms, Refs. [27,28,29,30] developed a more comprehensive real-time estimation algorithm for dynamic spatial and temporal wind fields in state-space representations using scalar measurements of a LiDAR. This allows for the easier integration of wind field preview for control algorithms of multiple turbines in a wind farm. The use of LiDAR s for aerospace research has steadily grown due to developments in the telecommunication industry, where components such as the optical fibres have allowed for their miniaturisation and improved reliability. Refs. [31,32] investigates the feasibility of the LiDAR based feedforward flight control system to minimise gust loads in high-speed civil transport and flexible aircraft respectively. Ref. [33] looks into both active gust load alleviation control and investigates the characterisation of aircraft wake vortices using airborne LiDAR systems. While LiDAR systems have become more and more compact and portable over the years, their size and weight are still too large for onboard use in most commercial UAVs. Tethered applications of LiDAR systems have been investigated using UAVs. Ref. [34] conducted a proof-of-concept of a tethered UAV-LiDAR system in which the LiDAR unit was kept on the ground and its telescopic transmitter-receiver was mounted to the UAV; The main application of which was to test the system as a wind-measurement platform. Ref. [35] simulates the use of an airborne LiDAR and a $H_{\infty }$ preview controller during the autonomous landing of a highly flexible fixed-wing aircraft to reject wind disturbances. However, the assumption that the LiDAR unit is kept onboard the aircraft wing limits the study to large scale aircraft and excludes most UAVs. In this paper a ground-based LiDAR unit is proposed to generate the preview needed for disturbance rejection during the takeoff and landing of a quadrotor UAV from its ground station. Here, the LiDAR unit has no physical connection to the autonomous UAV and hence does not limit its flight-envelope. This paper builds on previous work [36], where a prototype LiDAR unit was experimentally verified to predict the incoming wind disturbance at the landing site of UAV. The benefits of a feedforward control strategy were also assessed in this previous work. In the work presented here the disturbance rejection is improved further by incorporating the preview information received from the LiDAR in an MPC formulated for disturbance preview.

The main contributions of this paper include the development of a linear MPC with wind disturbance preview, the analysis of the MPC’s performance against varying preview lengths, the verification of the proposed performance benefits through non-linear simulations using experimental LiDAR data, and a preliminary analysis into the robustness of the system against uncertainty in the wind preview information. The remainder of this paper is organised as follows: The methodology, including system architecture, quadrotor modelling, wind preview MPC formulation, and description of the LiDAR-based wind preview system, are provided in Section 2. Results of simulation and analysis work with accompanying discussion are given in Section 3. Finally, Section 4 provides concluding remarks and highlights future work to be carried out.

## 2. Methods and Systems

### 2.1. Control Architecture

The architecture of the wind preview-based MPC is shown in Figure 1. As the LiDAR measurements are line-of-sight (LOS), a scanning procedure is employed to reconstruct the wind vector from its scalar measurements. This upstream wind vector is then fed into the wind predictor block which uses the quadrotor’s position within the landing site and a wind propagation model to predict its incoming disturbance. This wind preview is used by the MPC that sits in the outer loop of the aircraft’s overall control architecture and sends attitude demands to the inner loop. This cascading of controllers and the placement of the gust rejection control on the outer loop is for the following reasons:To impart modularity in the overall system and to allow the outer-loop control to be kept off-board the aircraft. This removes any weight constraints for the gust rejection controller and hence makes computational expensive controllers more feasible.To keep the inner-loop of the quadrotor unmodified and hence minimise the effect the outer-loop controller has on the stability characteristics of the aircraft. This can be further ensured by bounding the output attitude demands of the outer-loop controller.The identification of the aircraft model required for MPC design becomes easier. The placement of the controller on the outer-loop eliminates the need to identify/model the inner-loop dynamics of the aircraft which tend to be more non-linear. This also adds to the modularity of the gust rejection system as the model identification required for different UAVs becomes quicker.

### 2.2. Quadrotor Simulation Model

A mathematical model of the qaudrotor was developed for MPC design and simulation in this research. The equations of motion used for this model are the standard Newton-Euler equations, the translational component of which is given by:$$\vec{F_{b}}=\left[\begin{array}{c} F_{x}\\ F_{y}\\ F_{z} \end{array}\right]=m\left(\dot{\vec{V_{b}}}+\vec{\omega_{b}}\times \vec{V_{b}}\right)$$
where $\vec{F_{b}}$ is the total force applied on the rigid body in the aircraft’s body-fixed reference frame, *m* is the mass of the aircraft, $V_{b}$ is its velocity and $\vec{\omega_{b}}={\left[\begin{array}{c} p,q,r \end{array}\right]}^{⊺}$ are its rotational rates, both in the body-fixed reference frame. The rotational motion of the aircraft is defined by:$$\vec{M_{b}}=\left[\begin{array}{c} L\\ M\\ N \end{array}\right]=I\left(\dot{\vec{\omega_{b}}}+\vec{\omega_{b}}\times I\omega_{b}\right)$$
where $M_{b}$ is total rotational moment acting on the aircraft about its body-fixed reference frame; with its components ${\left[\begin{array}{c} L,M,N \end{array}\right]}^{⊺}$ corresponding to the total pitch, roll, and yaw moments respectively. *I* is the inertia matrix of the aircraft, assumed to be constant along with the mass of the aircraft. The translational velocity of the aircraft in the earth-fixed reference frame is obtained by the rotating the translational rates in the body-fixed reference frame using the direction cosine matrix,
$$R\left(\phi ,\theta ,\psi \right)=\left[\begin{array}{ccc} c_{\theta }c_{\psi } & c_{\theta }s_{\psi } & -s_{\theta }\\ s_{\phi }s_{\theta }c_{\psi }-c_{\phi }s_{\psi } & s_{\phi }s_{\theta }s_{\psi }+c_{\phi }c_{\psi } & s_{\phi }c_{\theta }\\ c_{\phi }s_{\theta }c_{\psi }+s_{\phi }s_{\psi } & c_{\phi }s_{\theta }s_{\psi }-s_{\phi }c_{\psi } & c_{\phi }c_{\theta } \end{array}\right]$$
where $\left[\begin{array}{c} \phi ,\theta ,\psi \end{array}\right]$ are the Euler/Tait-Bryan angles representing the orientation of the aircraft in the earth-fixed reference frame. Here $c_{\left(\cdot \right)}$ and $s_{\left(\cdot \right)}$ denote the cosine and sine of (·) respectively. Finally, the rotational rates in the earth-fixed reference frame can be obtained through the following rotation:$$\left[\begin{array}{c} \dot{\phi }\\ \dot{\theta }\\ \dot{\psi } \end{array}\right]=\left[\begin{array}{ccc} 1 & s_{\phi }t_{\theta } & c_{\phi }t_{\theta }\\ 0 & c\phi & -s_{\phi }\\ 0 & s_{\phi }/c_{\theta } & c_{\phi }/c_{\theta } \end{array}\right]\left[\begin{array}{c} p\\ q\\ r \end{array}\right]$$
where $t_{\left(\cdot \right)}$ denotes tan(·). The aerodynamics of the propeller and the airframe are integrated with the above equations of motion to allow the simulation of the aircraft model. To accurately capture the aircraft response under wind disturbances, the aerodynamics of each propeller is considered individually. The aerodynamic outputs of each propeller are given by
$$T=\rho A{\left(\Omega R\right)}^{2}C_{T}$$
$$H=\rho A{\left(\Omega R\right)}^{2}C_{H}$$
$$Q=\rho A\Omega^{2}R^{3}C_{Q}$$
where *T*, *H*, and *Q* are the propeller thrust, in-plane horizontal force, and torque respectively. ρ is the air density, *A* is the area of the propeller disc, Ω is the rotational speed of the propeller, *R* is its radius. $\left(C_{T},C_{H},C_{Q}\right)$ are the propeller coefficients for thrust, horizontal force, and torque respectively. These coefficients are a non-linear function of the aircraft’s airspeed, rotational speed of the propeller, and propeller blade characteristics. The expressions for the coefficients and mappings to various incidence angles, rotational speeds, and airspeeds are omitted here for brevity and since they are already detailed in previous related works, Refs. [37,38,39]. The airframe drag for the quadrotor is considered as follows:$$\vec{D_{b}}={\left[\begin{array}{c} D_{x}\\ D_{y}\\ D_{z} \end{array}\right]}_{b}=-\frac{1}{2}\rho C_{D}\vec{V_{b}}\cdot \vec{V_{b}}\left[\begin{array}{c} S_{x}\\ S_{y}\\ S_{z} \end{array}\right]$$
where $C_{D}$ is the airframe drag coefficient and ${\left[\begin{array}{c} S_{x},S_{y},S_{z} \end{array}\right]}^{⊺}$ is the surface area of the airframe along each respective axes. The above formulation simplifies the airframe drag and considers the aircraft as a cuboid and only accounts for form drag, ref. [40].

The Draganflyer X-Pro quadrotor, depicted in Figure 2, was used as the simulation aircraft in this research. The mass, inertia, and aerodynamic properties of which are provided in Table 1.

#### Linear Model for MPC Design

The aircraft simulation model described above contains non-linearities due to its aerodynamics and hence needs to be linearised for MPC design. The linearisation procedure was carried out by first trimming the aircraft at zero wind and at hover. The trimming and linearisation process were carried out numerically in MATLAB using the findOp and linmod commands respectively. The linearised model is obtained in a state-space representation with the following structure:$$\dot{\vec{x}}=A\vec{x}+B_{u}\vec{u}+B_{w}\vec{w}$$
$$\vec{y}=C\vec{x}$$
where *A* is the state-matrix, $B_{u}$ is the input matrix due to control input $\vec{u}$, $B_{w}$ is the wind effector matrix, $\vec{w}$ is the three-dimensional wind vector, $\vec{y}$ is the output vector, and *C* is the output matrix, and $\vec{x}$ is the state vector with the following definition:$$\vec{x}={\left[\begin{array}{cccccccccccc} x & y & z & u & v & w & \phi & \theta & \psi & p & q & r \end{array}\right]}^{⊺}$$
where $\left[\begin{array}{c} x,y,z \end{array}\right]$ is the position vector of the aircraft in the earth-fixed reference frame and $\left[\begin{array}{c} u,v,w \end{array}\right]=\vec{V_{b}}$, the translational velocity of the aircraft in the body-fixed reference frame. To obtain offset-free position tracking in wind, the attitude states of the aircraft were removed from the outputs, hence $\vec{y}$ is defined as:$$\vec{y}={\left[\begin{array}{ccccccccc} x & y & z & u & v & w & p & q & r \end{array}\right]}^{⊺}$$

This removal of attitude states in the outputs is necessary as the quadrotor holds position in wind with a non-zero attitude. Hence, the MPC should not try to track the attitude states of the quadrotor. It should be noted here that the attitude states are only removed from the tracked outputs of the MPC’s internal model and not its states. Full state-feedback is still required for the MPC’s prediction, detailed in Section 2.3. The state effector matrices *A*, $B_{u}$, and $B_{w}$ were obtained from the linearisation procedure as follows:
$$A={\left[\begin{array}{ccccccccccc} \; & 0 & 1 & 0 & 0 & 0 & 0 & 0 & 0 & 0 & 0\\ \; & 0 & 0 & 1 & 0 & 0 & 0 & 0 & 0 & 0 & 0\\ \; & 0 & 0 & 0 & 1 & 0 & 0 & 0 & 0 & 0 & 0\\ \; & 0 & X_{u} & 0 & 0 & 0 & -g & 0 & 0 & 0 & 0\\ \; & 0 & 0 & Y_{v} & 0 & g & 0 & 0 & 0 & 0 & 0\\ \cdots & Z_{z} & 0 & 0 & Z_{w} & 0 & 0 & 0 & 0 & 0 & 0\\ \; & \; & & \; & & \; & \vdots \\ \; & 0 & 0 & 0 & 0 & L_{\phi } & 0 & 0 & L_{p} & 0 & 0\\ \; & 0 & 0 & 0 & 0 & 0 & M_{\theta } & 0 & 0 & M_{q} & 0\\ \; & 0 & 0 & 0 & 0 & 0 & 0 & N_{\psi } & 0 & 0 & N_{r} \end{array}\right]}_{\left[12\times 12\right]}$$
$$B_{U}={\left[\begin{array}{cccc} \; & \;\;\;\;\;\;\;\;\;\;\;\vdots \\ Z_{u_{1}} & 0 & 0 & 0\\ \; & \;\;\;\;\;\;\;\;\;\;\;\vdots \\ 0 & 0 & L_{u_{3}} & 0\\ 0 & M_{u_{2}} & 0 & 0\\ 0 & 0 & 0 & L_{u_{4}} \end{array}\right]}_{\left[12\times 4\right]}B_{W}={\left[\begin{array}{ccc} 0 & 0 & 0\\ 0 & 0 & 0\\ 0 & 0 & 0\\ X_{u} & 0 & 0\\ 0 & Y_{v} & 0\\ 0 & 0 & Z_{w}\\ \; & \vdots \end{array}\right]}_{\left[12\times 3\right]}$$
where *g* is the acceleration due to gravity, $\left[X_{u},Y_{v},Z_{w}\right]$ are the aerodynamic drag derivatives in the respective axes, $Z_{z}$ is the vertical force derivative to height feedback, $\left[L_{\phi },M_{\theta },N_{\psi }\right]$ are the rotational moment derivatives due to attitude, $\left[L_{p},M_{q},N_{r}\right]$ are damping terms for the rotational moments, and $\left[Z_{u_{1}},M_{u_{2}},L_{u_{3}},N_{u_{4}}\right]$ are the control derivatives. The above input effector matrix, $B_{u}$, corresponds to a input vector $\vec{u}=\left[\begin{array}{c} u_{1},u_{2},u_{3},u_{4} \end{array}\right]$ where $u_{1}$ is the altitude control input, $u_{2}$ is the pitch angle control input, $u_{3}$ is roll angle control input and $u_{4}$ is the yaw angle control input. The values of these derivatives for the linearised model of the Draganflyer X-Pro are given in Table 2.

Figure 3 shows the architecture of the simulation environment. The control outputs from the MPC are then sent to the individual inner-loop proportional-integral-derivative (PID) controller to track the respective variable. The gains for these controller were determined using Simulink’s in-built PID tuner application and are given in Table 3. The error driven outputs of the PID controllers are $U_{1}$ the thrust collective, $U_{2}$ the pitch differential, $U_{3}$ the roll differential and $U_{4}$ the yaw differential. These control outputs are then set to a control allocation block that combines and translates each control input into a rotational speed demand for each propeller. The relation between the control output and propeller speed depends on the location of the propeller in the body reference frame of the aircraft. The model used here has been set for the ‘X’ configuration where the diametrically opposite propellers are placed along the forward and lateral body frame axes with propeller number 1 being placed along forward body axes and the remaining propellers following a clockwise numbering convention. The propeller speeds are then obtained using the following expressions:$$\vec{\Omega }=\left[\begin{array}{c} \Omega_{1}\\ \Omega_{2}\\ \Omega_{3}\\ \Omega_{4} \end{array}\right]=\left[\begin{array}{c} U_{1}+U_{2}+U_{3}+U_{4}\\ U_{1}+U_{2}-U_{3}-U_{4}\\ U_{1}-U_{2}-U_{3}+U_{4}\\ U_{1}-U_{2}+U_{3}-U_{4} \end{array}\right]$$
where $\Omega_{\left(i\right)}$ denotes the rotational speed of the (i)th propeller. The allocation of propeller speeds in this manner allows for the decoupling of the various control channels.

Figure 3 also depicts the use of the Kalman Filter (KF) to estimate the states of the quadrotor for feedback into the MPC. This was required as the access to the rotational rates from the onboard sensors was not possible for the intended aircraft for future flight tests. Furthermore, the use of a KF also allows for the minimization of errors caused by measurement and process noise.

### 2.3. MPC with Wind Disturbance Preview

MPC is a well-established control methodology that has found success in aerospace applications due to its ability to handle complex and nonlinear systems and optimize performance in real time whilst keeping the system within desired operating conditions. The basic principle of MPC is that by repeatedly solving an open-loop optimal control problem, closed-loop control can be applied. Following the measurement of the system state, the control sequence that minimizes some performance measures over a finite time horizon subject to a set of constraints is computed. The first control of the sequence is applied and at the next sampling instant, the process is repeated. Detailed descriptions can be found in standard reference texts [42,43,44].

The general design objective of an MPC is to compute a set of future control inputs for a system to optimize its future behaviour within a finite time horizon. This is defined as the prediction horizon of the MPC. To obtain an estimate of the future response of the system, MPC utilises a mathematical model of the system to be controlled. In this paper, the model detailed in Section 2.2 is used. The future response of the system is estimated by simulating this model forward in time in the length of the MPC’s prediction horizon. This prediction is then optimized over the same horizon according to a specific performance criterion and constraints, which are typically defined in terms of minimizing a cost function. In the standard feedback-only configuration, the MPC’s control optimization takes into account only the current system outputs to simulate forward in time, hence any future or current disturbance acting on the system needs to be sensed through its effects on the system outputs. This invariably adds a delay in the compensation from the MPC. If future disturbances can be measured or estimated then the MPC’s predictive methodology can be leveraged to account for the disturbance within its optimization scheme. This preview driven optimization translates to preemptive control inputs to compensate for the incoming/changing disturbance.

The formulation of a linear MPC with wind disturbance preview will now be discussed. This formulation is built on the standard formulation of MPC for discrete linear time-invariant systems [44], with the new addition of wind disturbance and wind preview into the MPC. Usually, wind acts as an unknown disturbance input into the system however if the incoming disturbance can be measured and estimated then this preview information can be used within the prediction horizon of the MPC to determine the optimal control inputs. For the discrete-time implementation of the MPC, we define the system as: (1)$$\begin{array}{c} \vec{x}\left(k+1\right)=A_{d}\vec{x}\left(k\right)+B_{d}\vec{u}\left(k\right)+B_{w}\vec{w}\left(k\right)\\ \vec{y}\left(k\right)=C_{d}\vec{x}\left(k\right) \end{array}$$
where $A_{d}$, $B_{d}$, $C_{d}$ are the discrete state-space matrices of the quadrotor system and $B_{w}$ is its wind disturbance matrix. $\vec{x}\left(k\right)$, $\vec{u}\left(k\right)$, $\vec{w}\left(k\right)$, and $\vec{y}\left(k\right)$ denote the state, input, wind disturbance and system output vectors at the $k^{th}$ timestep. Denoting the difference in state, output, and control input as follows:(2)$$\begin{array}{c} \Delta \vec{x}\left(k+1\right)=\vec{x}\left(k+1\right)-\vec{x}\left(k\right)\\ \Delta \vec{y}\left(k+1\right)=\vec{y}\left(k+1\right)-\vec{y}\left(k\right)\\ \Delta \vec{u}\left(k\right)=\vec{u}\left(k\right)-\vec{u}\left(k-1\right) \end{array}$$
and then substituting Equation (Equation 1) into the right hand side of Equation (Equation 2) we get,
(3)$$\begin{array}{c} \Delta \vec{x}\left(k+1\right)=A_{d}\Delta \vec{x}\left(k\right)+B_{d}\Delta \vec{u}\left(k\right)+B_{w}\Delta \vec{w}\left(k\right)\\ \Delta \vec{y}\left(k+1\right)=C_{d}A_{d}\Delta \vec{x}\left(k\right)+C_{d}B_{d}\Delta \vec{u}\left(k\right)+C_{d}B_{w}\Delta \vec{w}\left(k\right) \end{array}$$

Here $\Delta \vec{y}\left(k+1\right)$ denotes the difference in predicted output of the system at the next timestep (k+1) formulated in terms of the current state difference $\Delta \vec{x}\left(k\right)$ and the current control and disturbance input differences $\Delta \vec{u}\left(k\right)$ and $\Delta \vec{w}\left(k\right)$. Now defining the predicted state of the system at the ${\left(k+j\right)}^{th}$ timestep based on the current state $\vec{x}\left(k\right)$ as $\vec{x}\left(k+j|k\right)$, the state evolution can then be extended over the entire horizon of the MPC as:(4)$$\begin{array}{cc} \Delta \vec{x}(k+1\;|\;k)= & A_{d}\Delta \vec{x}\left(k\right)+B_{d}\Delta \vec{u}\left(k\right)+B_{w}\Delta \vec{w}\left(k\right)\\ \Delta \vec{x}(k+2\;|\;k)= & A_{d}\Delta \vec{x}\left(k+1\right)+B_{d}\Delta \vec{u}\left(k+1\right)+B_{w}\Delta \vec{w}\left(k+1\right)\\ = & A_{d}^{2}\Delta \vec{x}\left(k\right)+A_{d}B_{d}\Delta \vec{u}\left(k\right)+B_{d}\Delta \vec{u}\left(k+1\right)+A_{d}B_{w}\Delta \vec{w}\left(k\right)+B_{w}\Delta \vec{w}\left(k+1\right)\\ & \vdots \\ \Delta \vec{x}(k+N_{p}\;|\;k)= & A_{d}^{N_{p}}\Delta \vec{x}\left(k\right)+A_{d}^{N_{p}-1}B_{d}\Delta \vec{u}\left(k\right)+A_{d}^{N_{p}-2}B_{d}\Delta \vec{u}\left(k+1\right)\\ & +A_{d}^{N_{p}-1}B_{w}\Delta \vec{w}\left(k\right)+A_{d}^{N_{p}-2}B_{w}\Delta \vec{w}\left(k+1\right)\\ & \cdots +A_{d}^{N_{p}-N_{c}}B_{d}\Delta \vec{u}\left(k+N_{c}-1\right)+A_{d}^{N_{p}-N_{c}}B_{w}\Delta \vec{w}\left(k+N_{c}-1\right) \end{array}$$
where $N_{p}$ and $N_{c}$ are the prediction and control horizons respectively. Now from the predicted states we can obtain the predicted outputs through substitution:(5)$$\begin{array}{cc} \Delta \vec{y}\left(k+1\;|\;k\right)= & C_{d}A_{d}\Delta \vec{x}\left(k\right)+C_{d}B_{d}\Delta \vec{u}\left(k\right)+C_{d}B_{w}\Delta \vec{w}\left(k\right)\\ \Delta \vec{y}\left(k+2\;|\;k\right)= & C_{d}A_{d}^{2}\Delta \vec{x}\left(k\right)+C_{d}A_{d}B_{d}\Delta \vec{u}\left(k\right)+C_{d}B_{d}\Delta \vec{u}\left(k+1\right)\\ & +C_{d}A_{d}B_{w}\Delta \vec{w}\left(k\right)+C_{d}B_{w}\Delta \vec{w}\left(k+1\right)\\ & \vdots \\ \Delta \vec{y}\left(k+N_{p}\;|\;k\right)= & C_{d}A_{d}^{N_{p}}\Delta \vec{x}\left(k\right)+C_{d}A_{d}^{N_{p}-1}B_{d}\Delta \vec{u}\left(k\right)+C_{d}A_{d}^{N_{p}-2}B_{d}\Delta \vec{u}\left(k+1\right)\\ \; & +C_{d}A_{d}^{N_{p}-1}B_{w}\Delta \vec{w}\left(k\right)+C_{d}A_{d}^{N_{p}-2}B_{w}\Delta \vec{w}\left(k+1\right)\\ \; & \cdots +C_{d}A_{d}^{N_{p}-N_{c}}B_{d}\Delta \vec{u}\left(k+N_{c}-1\right)+C_{d}A_{d}^{N_{p}-N_{c}}B_{w}\Delta \vec{w}\left(k+N_{c}-1\right) \end{array}$$
Hence all predicted outputs in the above equation are in terms of the current state x(k), the future control inputs $\Delta \vec{u}\left(k+j\right)$ and future wind inputs $\Delta \vec{w}\left(k+j\right)$ where $j=0,1,\cdots N_{c}-1$. Defining vectors for the future outputs, control inputs and wind inputs:$$\Delta Y={\left[\begin{array}{cc} \Delta \vec{y}\left(k+1\;|\;k\right) & \Delta \vec{y}\left(k+2\;|\;k\right)\cdots \Delta \vec{y}\left(k+N_{p}\;|\;k\right) \end{array}\right]}^{T}$$
$$\Delta U={\left[\begin{array}{cc} \Delta \vec{u}\left(k\right) & \Delta \vec{u}\left(k+2\right)\cdots \Delta \vec{u}\left(k+N_{c}-1\right) \end{array}\right]}^{T}$$
$$\Delta W={\left[\begin{array}{cc} \Delta \vec{w}\left(k\right) & \Delta \vec{w}\left(k+2\right)\cdots \Delta \vec{w}\left(k+N_{c}-1\right) \end{array}\right]}^{T}$$

Hence the predicted outputs can be written in the compacted matrix form as:(6)$$\Delta Y=F\Delta \vec{x}\left(k\right)+\Phi_{U}\Delta U+\Phi_{W}\Delta W$$
where,
(7)$$F=\left[\begin{array}{c} C_{d}A_{d}\\ C_{d}A_{d}^{2}\\ C_{d}A_{d}^{3}\\ \;\;\vdots \\ C_{d}A_{d}^{N_{p}} \end{array}\right];\Phi_{U}=\left[\begin{array}{ccccc} C_{d}B_{d} & 0 & 0 & \cdots & 0\\ C_{d}A_{d}B_{d} & C_{d}B_{d} & 0 & \cdots & 0\\ C_{d}A_{d}^{2}B_{d} & C_{d}A_{d}B_{d} & C_{d}B_{d} & \cdots & 0\\ \;\;\vdots \\ C_{d}A_{d}^{N_{p}-1}B_{d} & C_{d}A_{d}^{N_{p}-2}B_{d} & C_{d}A_{d}^{N_{p}-3}B_{d} & \cdots & C_{d}A_{d}^{N_{p}-N_{c}}B_{d} \end{array}\right]$$
(8)$$\begin{array}{c} \Phi_{W}=\left[\begin{array}{ccccc} C_{d}B_{w} & 0 & 0 & \cdots & 0\\ C_{d}A_{d}B_{w} & C_{d}B_{w} & 0 & \cdots & 0\\ C_{d}A_{d}^{2}B_{w} & C_{d}A_{d}B_{w} & C_{d}B_{w} & \cdots & 0\\ \;\;\vdots \\ C_{d}A_{d}^{N_{p}-1}B_{w} & C_{d}A_{d}^{N_{p}-2}B_{w} & C_{d}A_{d}^{N_{p}-3}B_{w} & \cdots & C_{d}A_{d}^{N_{p}-N_{c}}B_{w} \end{array}\right] \end{array}$$

This predictive model can now be used along with the wind preview and future control inputs to optimise the future response of the system. For this optimization we define the cost function as:(9)$$J={\left(R_{s}-\Delta Y\right)}^{T}\left(R_{s}-\Delta Y\right)+\Delta U^{T}\bar{R}\Delta U$$
where $R_{s}$ is the vector containing the reference information and $\bar{R}$ is a diagonal weighting matrix to impart cost on high control efforts. The diagonal terms in $\bar{R}$ can be modified to tune the closed-loop performance of the system. Substituting Equations (Equation 6)–(Equation 8) into the cost function and taking the partial derivative w.r.t ΔU gives:(10)$$\begin{array}{c} \begin{array}{cc} \frac{\partial J}{\partial \Delta U}= & -2\Phi^{T}\left(R_{s}-F\Delta \vec{x}\left(k\right)\right)+2\left(\Phi^{T}\Phi +\bar{R}\right)\Delta U\\ \; & +\Phi^{T}\Phi_{d}\Delta W+\Delta W^{T}\Phi_{d}\Phi \end{array} \end{array}$$

The minimum of the cost function *J* is obtained by equating the above equation to zero, which gives the optimal set of control inputs ΔU as:(11)$$\begin{array}{c} \begin{array}{cc} \Delta U= & {\left(\Phi_{U}^{T}\Phi_{U}+\bar{R}\right)}^{-1}\left(\Phi_{U}^{T}\left(R_{s}-F\Delta \vec{x}\left(k\right)\right)-\frac{1}{2}\left(\Phi_{U}^{T}\Phi_{W}\Delta W+\Delta W^{T}\Phi_{W}^{T}\Phi_{U}\right)\right) \end{array} \end{array}$$

For the receding horizon control strategy employed in MPC, only the first input of optimised ΔU is sent through to the quadrotor. The prediction and optimisation is then repeated at the next timestep with the updated state of the system.
$$\Delta \vec{u}\left(k\right)=\left[\begin{array}{ccccc} 1 & 0 & 0 & \cdots & 0 \end{array}\right]\Delta U$$

Hence the control input into the system is then obtained as:$$\vec{u}\left(k\right)=\vec{u}\left(k-1\right)+\Delta \vec{u}\left(k\right)$$

#### Constraint Handling for Real-Time Implementation

In MPC methodology, constraints on the system are defined as linear inequalities w.r.t to the decision variable ΔU.
MΔU≤γ
where *M* is a matrix defining how the constraints relate to ΔU and γ is a vector containing the numeric limits of the constraints. The MPC optimisation now becomes a quadratic programming problem where the goal is to find the constrained minimum of a positive definite quadratic cost function subject to linear inequality constraints. In this paper, Hildreth’s quadratic programming algorithm [45], is used to numerically solve the constrained optimisation. The strength of Hildreth’s programming is its ability to find near-optimal solutions during conflicting constraints. Hence, it can automatically recover form ill-conditioned constraint problems. This attribute is key to the safe real-time operation of MPC in flight control systems of UAVs. Detailed implementation of the algorithm in MATLAB is provided in [44].

### 2.4. KF for State Estimation

Due to practical considerations with the intended test aircraft, rotational rate measurements are not available for direct feedback. Hence a discrete-time linear KF was used to estimate the missing states using the available measurements and a linear model of the aircraft. The same linear model and discretisation step used in MPC design is used here. The KF follows a standard formulation, the structure of which is summarised in the following equations: (12)$$\hat{\vec{x}}\left(k|k-1\right)=A_{d}\hat{\vec{x}}\left(k-1|k-1\right)+B_{d}\vec{u}\left(k\right)+B_{w}\vec{w}\left(k\right)$$
(13)$$P\left(k|k-1\right)=A_{d}P\left(k-1|k-1\right){A_{d}}^{⊺}+Q$$
(14)$$K\left(k\right)=\frac{P\left(k|k-1\right)H^{⊺}}{\left(HP\left(k|k-1\right)H^{⊺}+R\right)}$$
(15)$$\hat{\vec{x}}\left(k|k\right)=\hat{\vec{x}}\left(k|k-1\right)+K\left(k\right)\left(z\left(k\right)-H\hat{\vec{x}}\left(k|k-1\right)\right)$$
(16)P(k|k)=(I−K(k)H)P(k|k−1)
(17)$$\hat{\vec{x}}\left(k+1|k\right)=\hat{\vec{x}}\left(k|k\right)$$
(18)P(k+1|k)=P(k|k)

Equations (Equation 12) and (Equation 13) represent the KF prediction at current timestep *k* from the previous state estimate $\hat{\vec{x}}\left(k-1|k-1\right)$, and previous error covariance matrix P(k−1|k−1). Here $\hat{\vec{x}}\left(k|k-1\right)$ and P(k|k−1) represent the *a priori* estimates for the state vector and error covariance matrix at the current timestep and *Q* is the process noise covariance matrix. Equation (Equation 14) represents the calculation of the Kalman gain K(k), for the current timestep using the *a priori* error covariance matrix estimate, the output matrix of KF model *H*, and the sensor noise covariance *R*. Equations (Equation 15) and (Equation 16) represent the update step of the KF algorithm which yields the state estimate and error covariance matrix for the current timestep, using the *a priori* estimates and current measurement vector, z(k). This state estimate is then sent to the MPC for its control calculation. Finally, Equations (Equation 17) and (Equation 18) represent the projection step of the KF algorithm where the current estimates are projected into the next timestep for the subsequent calculations. Here, $\hat{\vec{x}}\left(k+1|k\right)$ and P(k+1|k) are known as the *a posteriori* state estimate and error covariance matrix respectively. As the rotational rates are not directly measured, the measurement vector takes the form:$$\vec{z}={\left[\begin{array}{ccccccccc} x & y & z & u & v & w & \phi & \theta & \psi \end{array}\right]}^{⊺}$$

The KF was then configured to estimate the rotational rates using the above measurements by setting its model output matrix, *H*, to disregard the rotational rates. This is possible as the system model described by matrices $A_{d},B_{d},B_{w}$ and *H* is still observable. The covariance matrices for process noise and sensor noise were fixed at 0.1 and 0.3 respectively in this work. The sensor noise covariance was obtained by using historical data of the quadrotor’s sensor measurements while the process noise covariance was obtained heuristically in simulation by looking at the estimation accuracy of the unmeasured states for different process noise covariance values while keeping the sensor noise covariance fixed.

### 2.5. LiDAR for Wind Preview

To measure the incoming wind disturbance an atmospheric wind LiDAR will be used. These LiDAR s use the principle of Doppler effect to estimate the speed of air particles by measuring the Doppler shift in frequency of the back-scattered laser. The LiDAR unit investigated in this research is a prototype version of a miniaturised continuous wave (CW) coherent LiDAR system that utilises the infra-red band for measurement. The wind-speed measurement range of the unit exceeds the flight-envelope of the test vehicle and most commercial quadrotor UAVs. The unit is a class 3 b laser that weighs less than 5 kg and its approximate physical dimensions are 35 cm × 25 cm × 12 cm. Detailed specification of this LiDAR unit is omitted here due to commercial sensitivity.

As the LiDAR unit measures LOS wind speed, a scanning procedure needs to be used along side its use for wind vector reconstruction. A mechanical mount was devised to horizontally scan the laser beam to reconstruct the horizontal wind vector. Once the upstream wind vector is reconstructed a wind propagation model is used to predict the downstream wind disturbance and hence provide the wind preview. Figure 4 depicts the mechanical mount used for the LiDAR scan procedure and illustrates the system architecture of the proposed MPC based gust rejection control system including the various system components used for wind preview generation. Detailed explanation of the various system nodes, methods for wind reconstruction, wind prediction, and their experimental verification has been conducted in previous work [36]. The wind data gathered from these experiments has been used in this paper to allow for more representative simulations.

## 3. Results and Discussion

This section looks at the results of the simulations conducted using the aircraft model described in Section 2.2 and the MPC designed using the same. A sample time of 0.1 s was used for the discretisation and the MPC’s weighting matrix used was kept as identity for all simulations. Section 3.1 will first look at simulations of the linearised model of the quadrotor with wind-preview-based MPC control, highlighting the performance improvements over an MPC with no disturbance preview. Section 3.2 shows the analysis results of varying preview length in the MPC. Remaining subsections utilise the non-linear model of the quadrotor with the addition of measurement noise, non-linear aerodynamics and experimental wind data; thus allowing for more representative results. Section 3.3 looks at the estimation results of the KF with added noise and the complete system performance, MPC + KF, in the presence of gusts. Section 3.3 looks at simulations using the experimental wind data gathered from the LiDAR as the wind preview for the MPC. Finally, Section 3.5 finishes with a analysis into the robustness of the wind-preview MPC based on system performance in the presence of gusts with uncertainty. MPC’s control and prediction horizons were kept fixed at 1 s in all simulations except for those in Section 3.2 where the MPC’s horizons were changed according to the wind preview length considered.

### 3.1. Linear Model Simulations–MPC with Wind Preview

Figure 5 shows the linear system response to a forward wind disturbance with and without wind-preview compensation in the MPC. The applied wind disturbance is a step input of 10 ms^−1^ at time t=2 s. A significant reduction in the quadrotor’s position deviation is observed with wind preview. There are also significant reductions in peak pitch rate, pitch angle, and control input magnitude with wind preview. These improvements are accredited to the MPC’s preemptive control action in pitch before the arrival of the gust as seen in Figure 5b. This preemptive pitch input start as soon as the gust enters the prediction horizon of the MPC, the window length of which was set to 1 s for this simulation.

Figure 6 plots the MPC’s internal prediction of the aircraft’s forward position at time t=1.5 s for the same gust starting at t=2 s compared against the actual response in aircraft position for the same time range. From Figure 6 we can see that with the addition of the wind disturbance matrix along with wind preview has allowed the MPC to better predict the aircraft’s trajectory in wind. Hence it is clear that the MPC’s optimisation scheme has deliberately moved the aircraft forward preemptively to minimise its position deviation over the prediction horizon defined. It should also be noted that this optimization accounts for the system lag in reaching the desired pitch attitude due to the model based prediction in the MPC. The following subsection will look into how varying the window length of the MPC’s prediction horizon will effect the system response.

### 3.2. Linear Simulations–Prediction Horizon Length

Figure 7 plots the linear system response to a forward step input in wind (at time t=10 s) for increasing prediction horizon lengths in the MPC, from $T_{p}=1$ s to $T_{p}=10$ s. It is assumed here that the length of the wind preview increases alongside the MPC’s prediction horizon. From the responses we can see that the most noticeable difference with increasing prediction length is in the position response of the aircraft, albeit the differences are in the cm scale. We can observe that increasing the horizon beyond 1 s also commences an initial backward motion of the aircraft before its forward motion to counter the backward push of the incoming wind. This two-stage preemptive motion of the aircraft results in a lower peak magnitude in position deviation to either side of the position set-point and thus results in a lower variance of the position response of the aircraft. This multi directional preemptive motion of the aircraft is a resultant of the MPC optimisation scheme which aims to minimise the position error of the aircraft from its set-point over the defined prediction horizon. Hence, by first moving the aircraft backwards and then forwards against the headwind, the system can build up momentum to counter the sudden push from the wind while still minimising its position deviation throughout the length of the horizon. The magnitude of this initial backward motion increases with increasing horizon length until $T_{p}=5$ s. Increasing the preview length further results in a three-stage preemptive movement of the aircraft where an initial forward movement is followed by a backward motion and then finishing with a forward motion before the wind impacts the aircraft. Doubling the preview length to $T_{p}=10$ s showed no noticeable difference to the system response when compared to $T_{p}=5$ s.

Table 4 lists the Root-mean-square (rms) values of the aircraft states with and without preview as well as with varying lengths of wind preview in the MPC. Correlating the forward position response of the aircraft in Figure 7 with Table 4, it is clear that the multi-stage preemptive motion of the aircraft for increasing preview lengths has in fact reduced the rms value of the forward position error.

### 3.3. Non-Linear Model Simulations

All results within this subsection were obtained from simulations of the non-linear aircraft model. To reduce the influence of measurement noise in the system performance with MPC, a KF was used to estimate the aircraft states from the noisy measurements. Figure 8 shows the KF estimation against the true states during a step demand in forward position. White noise with a covariance of 0.3 was added to all measured outputs. Due to practical constraints of the intended test aircraft, the rotational rates of the aircraft were not available for measurement. Hence, the KF was configured to estimate the rotation rates needed by the MPC. Figure 8d shows the estimation of the unmeasured rotational rate during the step input.

Figure 9 plots the system response when the aircraft is subjected to a forward gust and the KF estimates are used for MPC feedback. To better understand the benefits of the preview based control, a more realistic gust profile was used in this simulation, see Figure 9a. This gust profile consists of more representative rates in the windspeed variation. The improvement in position holding performance with wind preview is subdued in the non-linear simulation compared to the linear simulation but still significant. The addition of measurement noise and the non-linear aerodynamics in these simulations has contributed to the reduction in performance improvement observed with the linear MPC. However, a 66% improvement in position holding performance is still observed with the addition of wind preview; The aircraft is able to stay within ±20 cm with wind preview compared to ±60 cm without wind preview. The peaks in pitch rate, pitch angle and control input are also slightly reduced with wind preview, however their reductions are not as large as in the linear simulations. This reduction in improvement is mainly due to the more gradual profile of the gust unlike the step disturbance used in the linear case. This highlights that preview based control becomes more beneficial as the disturbance becomes more abrupt or sudden.

### 3.4. Non-Linear Simulations–MPC with LiDAR Data

In the simulations conducted here, the experimental data set gathered from the LiDAR tests conducted in previous work [36] was used. Here, the prediction estimates obtained from the upstream LiDAR measurements were used as the preview input into the MPC while the actual downstream measurement of the corresponding data set was used as the wind distance input into the quadrotor model. This in turn adds the prediction errors and measurement noise of the LiDAR to the MPC, thus allowing for a more representative simulation. Figure 10 shows the histograms of the aircraft’s response for the non-linear simulation. The performance improvements in this scenario have diminished when compared with the ideal linear simulations but are still significant. Much like the previous Subsection’s simulations, the position hold improvement is the most significant while the improvements in rotational rates, peak attitude and control signals have been subdued. While these simulations indicate that the wind-preview MPC is robust towards real word wind prediction data, the following Subsection will quantify the robustness further.

### 3.5. Robustness Analysis

System robustness was tested by adding increasing amount of structured uncertainty into the wind preview information used by the MPC in non-linear model simulations. The uncertainty is added into two parameters of the preview information, the magnitude of the incoming gust, $\Delta_{g}\left(\%\right)$, and its timing, $\Delta_{t}$ (s). Figure 11 shows how each of the parameterised uncertainties is applied on wind preview information. Here system robustness is quantitatively defined as the acceptable margins in either gust strength or timing before the system performance becomes similar to that without wind preview. i.e. the addition of wind preview even with erroneous magnitude and timing has not deteriorated the performance to worse than without the preview. Table 5 shows the results of the uncertainty analysis through the rms values of key system outputs with increasing uncertainty either in the strength of the gust or in its timing. From the analysis we can see that the system can handle up to approx. 120% uncertainty in the magnitude of the preview or approx. 1.75 s uncertainty in its timing before the system performance becomes lower than without wind preview. These results add to the practical feasibility of the wind-preview based MPC. The accuracy of the experimental data gathered in [36] fall well within the margins calculated in the simulations here.

## 4. Summary and Conclusions

In this paper wind preview-based Model Predictive Control is proposed to increase the gust resilience of multi-rotor UAVs. A linear MPC with disturbance preview capability was formulated using a mathematical model of a quadrotor. Remote wind sensing instruments such as atmospheric LiDAR can be used to predict the wind profile needed for the wind preview-based MPC. The preview information was used within the MPC to provide optimal disturbance rejection over its prediction horizon. The proposed control architecture places the wind-preview-based MPC off-board the UAV and in the outer-loop attitude channel to eliminate weight and hence computational constraints of the MPC and to minimise the impact on aircraft stability. Simulations were conducted using experimental LiDAR data to assess the performance benefits with the addition of wind preview. Analyses on wind preview length and system robustness were also conducted through simulation.

It was shown that wind preview-based MPC can significantly reduce the position deviation of the UAV under gust and thus improve its tracking performance. Reductions were also observed in translational, rotational rates, and maximum aircraft attitude during the disturbance compensation. This allows for a less aggressive approach to increased gust rejection. This key attribute of wind-preview-based MPC control differentiates it from conventional feedback-based methods where delays in measurement and actuation have to be dealt with larger control signals to achieve the same performance. Gust simulations with varying prediction horizon lengths demonstrate the preemptive nature of the wind-preview-based MPC. System robustness was tested in simulation by adding structured uncertainty into the preview information used by the MPC; a 120% or 1.75-s margin was possible in either the preview’s magnitude or timing before system performance dropped below that without wind preview.

Although wind-preview MPC provides improved disturbance rejection with lower magnitude control signals, aircraft attitudes, and rates, it requires additional instrumentation and increased computation. Hence it should be used for applications where gust resilience is safety critical. This attribute could prove advantageous for UAVs in outdoor ‘drone-delivery’ applications where aircraft and hence payload orientation are critical for stability and operational constraints. Moreover, electric VTOL aircraft could also leverage this less aggressive control technique to improve passenger comfort during takeoff and landing from gusty environments such as the rooftops of high-rise buildings. Additionally, wind preview-based flight control of UAVs can also reduce the time required to complete tasks where precise position control is required; such as the docking of the aircraft with its charging station. This is again possible due to the preemptive control nature of preview-based systems that yield reduced settling time in the presence of disturbances.

### Future Work

The next steps for this research include the experimental validation of the proposed wind-preview-based MPC. System identification tests will be carried out beforehand to obtain the model of the test aircraft that will be used for MPC implementation. The flight test platform and intended aircraft will be the ones used in previous work [36]. Integration of the wind-preview-based MPC into a trajectory planner will also be carried out. This will potentially aid the obstacle avoidance of autonomous UAVs during close proximity inspections in gusty environments. The comparison of preview-based MPC control against other pure feedback-based disturbance attenuation methods is another avenue for future work that will be explored.

## Figures and Tables

**Figure 1 sensors-23-03711-f001:**
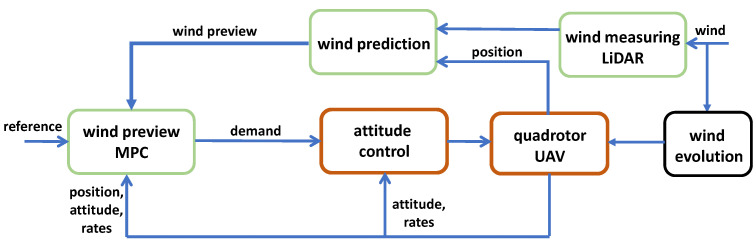
Overall system architecture of the gust rejection control system for the quadrotor UAV.

**Figure 2 sensors-23-03711-f002:**
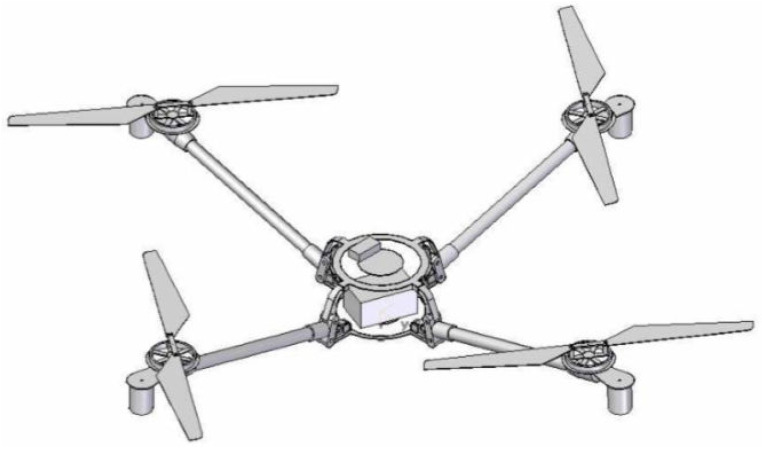
Draganflyer X-Pro quadrotor used for simulation, ref. [41].

**Figure 3 sensors-23-03711-f003:**
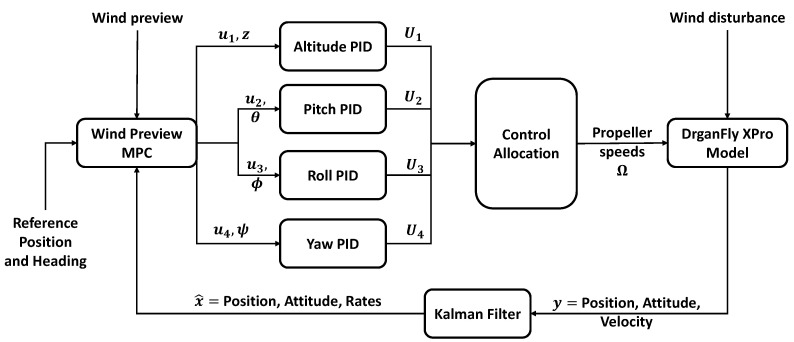
Simulation model architecture. MPC outputs and inputs into plant model are separated to allow for decoupled control action.

**Figure 4 sensors-23-03711-f004:**
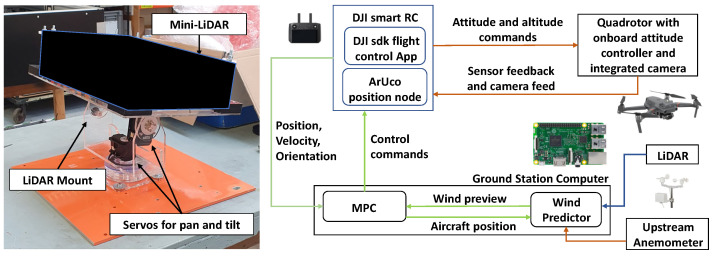
LiDAR on its scanning mount (**left**) and the proposed Quadrotor flight-test platform architecture (**right**).

**Figure 5 sensors-23-03711-f005:**
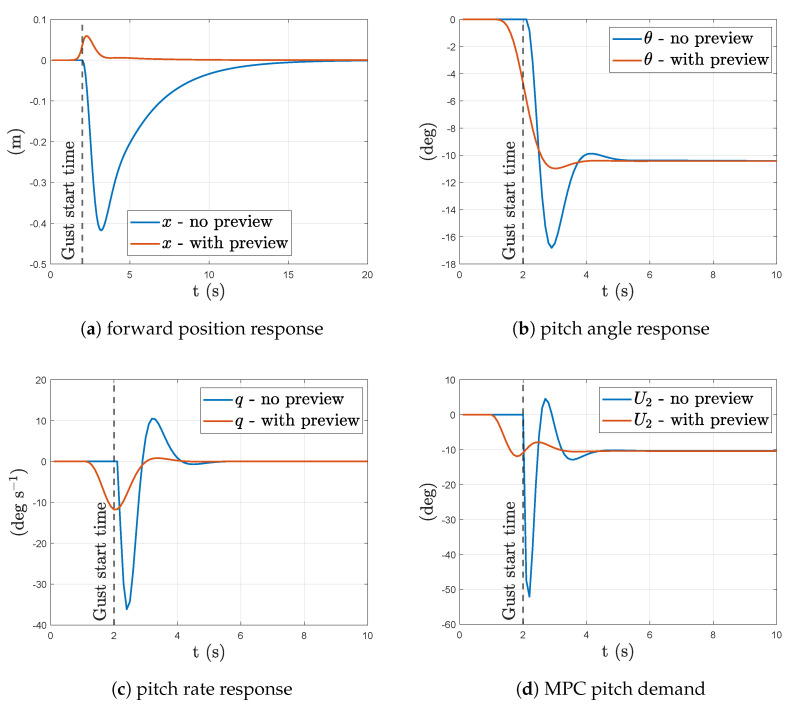
Linear aircraft response with and without wind preview control to a step input in forward wind of 10 ms^−1^ at time t=2 s.

**Figure 6 sensors-23-03711-f006:**
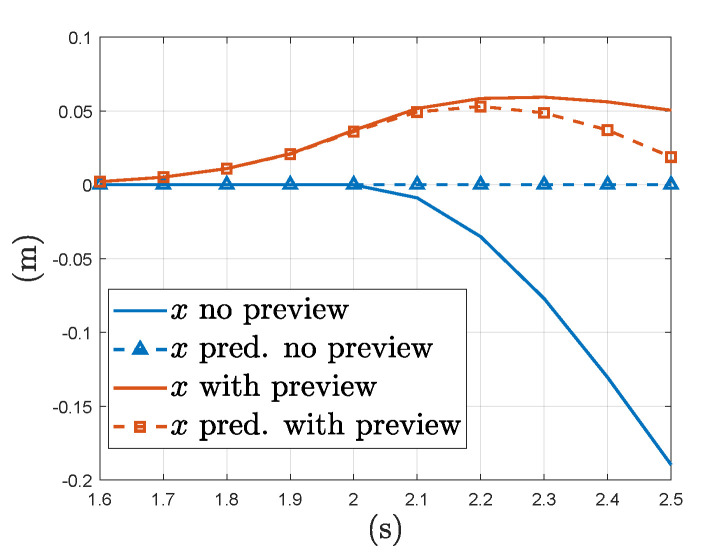
MPC forward position prediction vs actual response with a step input in forward wind of 10 ms^−1^ at time t=2 s.

**Figure 7 sensors-23-03711-f007:**
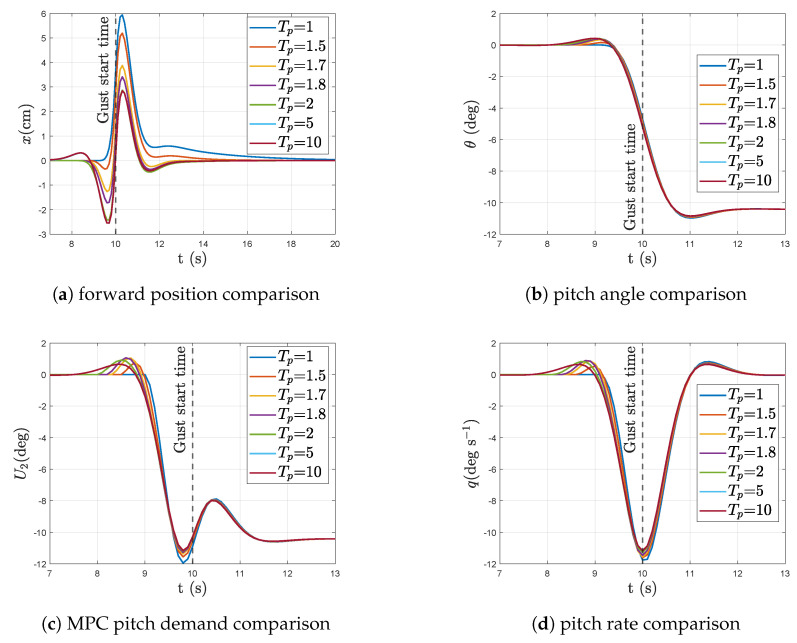
Linear aircraft response with varying wind preview lengths to a step input in forward wind of 10 ms^−1^ at time t=10 s.

**Figure 8 sensors-23-03711-f008:**
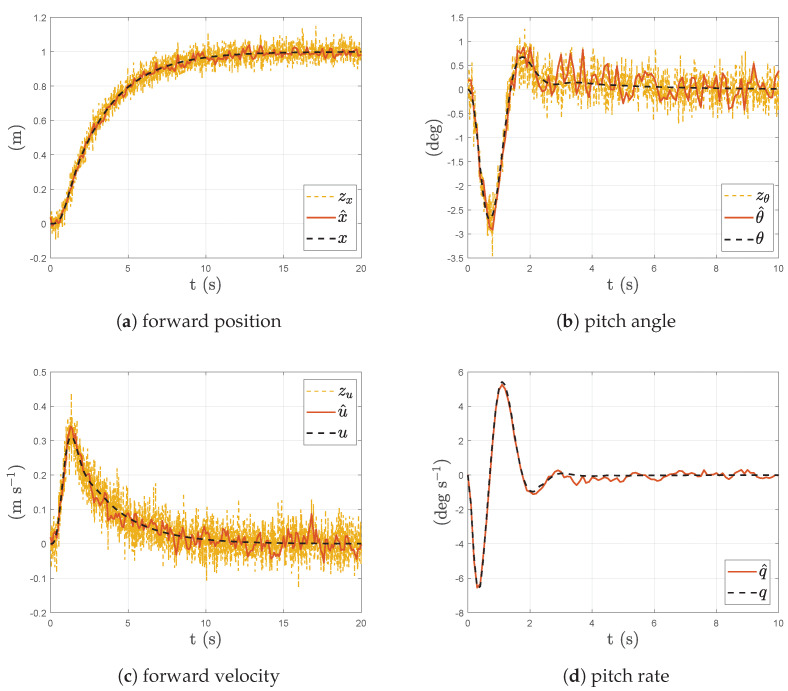
KF estimates (KF not in-the-loop) compared with measurements and true state during step response in position of non-linear aircraft model.

**Figure 9 sensors-23-03711-f009:**
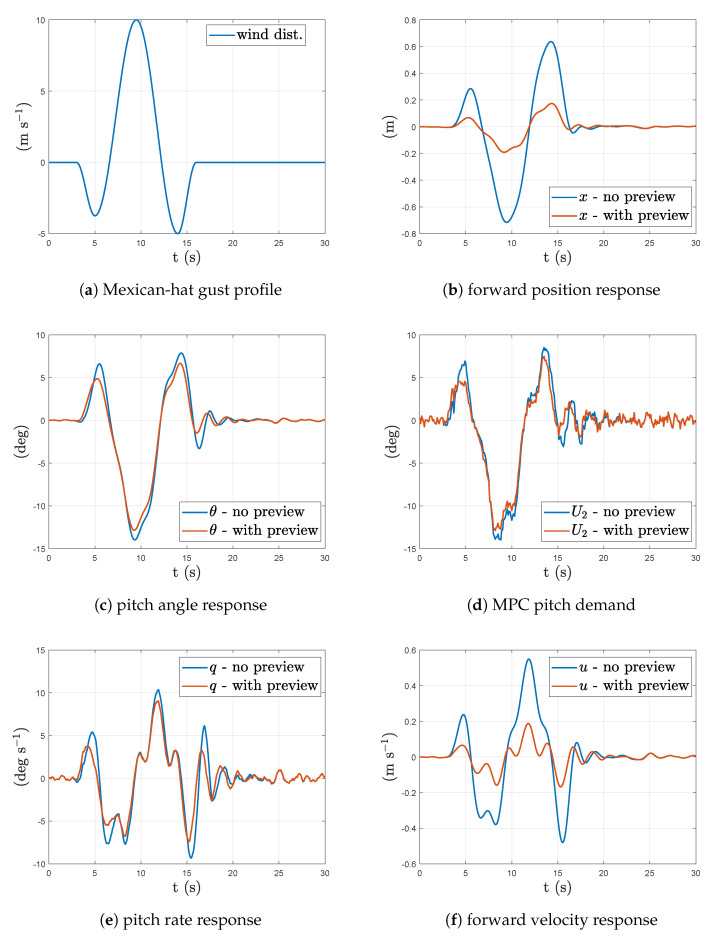
Non-linear aircraft simulation response with and without wind preview control to a Mexican-hat shaped gust profile. One second wind preview.

**Figure 10 sensors-23-03711-f010:**
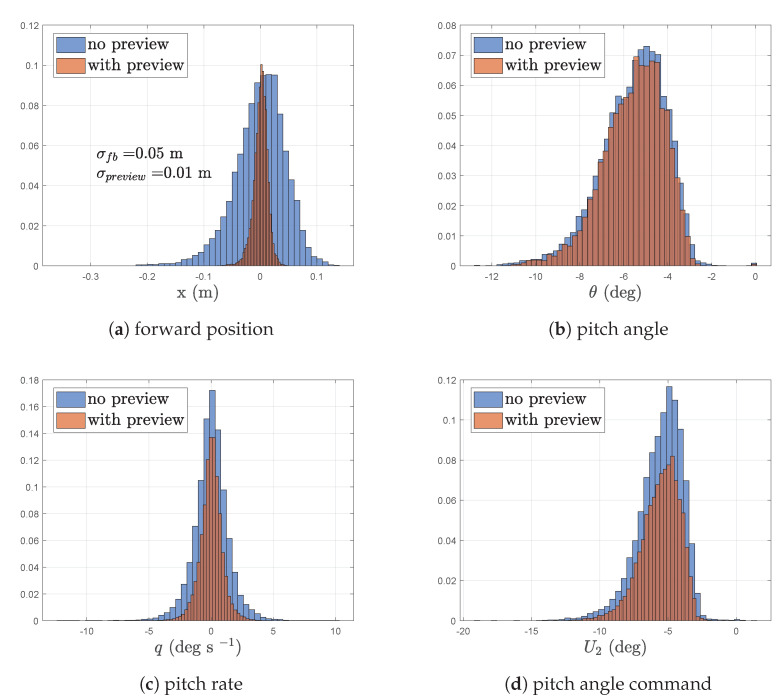
Histograms with and without wind preview control when aircraft is subjected to experimental wind data. Vertical axis for all plots is probability density.

**Figure 11 sensors-23-03711-f011:**
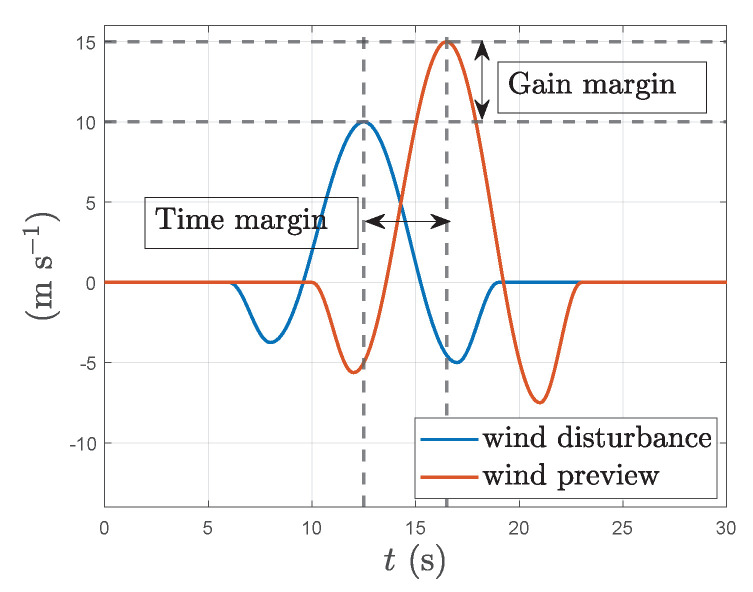
Representations of gust magnitude and time uncertainty used in robustness analysis.

**Table 1 sensors-23-03711-t001:** Mass, inertia, and aerodynamic properties used to build mathematical model of the Drganflyer XPro quadrotor, ref. [41]. diag(·) denotes a diagonal matrix.

Property	Value
Mass (kg)	2.36
Inertia matrix (kg m^2^)	$I_{b}=\mathrm{diag}\left(0.1676,0.1686,0.2974\right)$
Surface areas $\left[S_{x},S_{y},S_{z}\right]$ (m^2^)	$\left[\begin{array}{ccc} 0.0076 & 0.018 & 0.0195 \end{array}\right]$
Airframe drag coeff. ($C_{D}$)	1.05
Propeller arm length (m)	0.45
Number of blades per propeller	2
Blade chord (m)	0.04
Blade lift-curve-slope	5.5
Blade pitch (rad)	0.3025
Blade zero-lift drag coeff. ${\left(C_{D_{0}}\right)}_{b}$	0.05
Blade drag coeff. due to angle of attack ${\left(C_{D_{\alpha }}\right)}_{b}$	0.7

**Table 2 sensors-23-03711-t002:** Aerodynamic and control derivative values of the linear Draganflyer X-Pro model obtained through numerical linearisation.

Aerodynamic/Control Derivative	Value
Forward force due to forward airpseed—$X_{u}$ (N s m^−1^)	−0.1785
lateral force due to lateral airpseed—$Y_{v}$ (N s m^−1^)	−0.1785
Vertical force due to vertical airpseed—$Z_{w}$ (N s m^−1^)	−1.2495
Vertical force due to altitude—$Z_{z}$ (N m^−1^)	−0.8169
Rolling moment due to roll angle—$L_{\phi }$ (N m rad^−1^)	−3.6594
Pitching moment due to pitch angle—$M_{\theta }$ (N m rad^−1^)	−3.6376
Yawing moment due to yaw angle—$N_{\psi }$ (N m rad^−1^)	−0.0785
Rolling moment due to roll rate—$L_{p}$ (N m s rad^−1^)	−1.7815
Pitching moment due to pitch rate—$M_{q}$ (N m s rad^−1^)	−1.7709
Yawing moment due to yaw rate—$N_{r}$ (N m s rad^−1^)	−0.2868
Vertical force due to control input $u_{1}$—$Z_{u_{1}}$ (N)	−0.8160
Pitching moment due to control input $u_{2}$—$M_{u_{2}}$ (N m)	3.6376
Rolling moment due to control input $u_{3}$—$L_{u_{3}}$ (N m)	3.6594
Yawing moment due to control input $u_{4}$—$N_{u_{4}}$ (N m)	0.0785

**Table 3 sensors-23-03711-t003:** PID gains of the different inner-loop controller for the quadrotor simulation model.

Control Channel	Proportional Gain	Integral Gain	Derivative Gain
Altitude	5.89	0.80	8.72
Pitch/roll	5.89	0.13	1.62
Yaw	2.44	0.08	6.56

**Table 4 sensors-23-03711-t004:** Root-mean-square (rms) values of position, MPC pitch demand, pitch response, and pitch rate for different wind preview lengths.

Wind Preview Length (s)	$x_{\mathrm{rms}}$ (cm)	${\left(U_{2}\right)}_{\mathrm{rms}}$ (deg)	$\theta_{\mathrm{rms}}$ (deg)	$q_{\mathrm{rms}}$ (deg s^−1^)
No preview	14.57	10.58	8.71	5.9
$T_{p}$ = 1.0	1.25 (−91%)	8.66 (−18%)	8.44 (−3%)	2.51 (−57%)
$T_{p}$ = 1.5	1.04 (−16%)	8.66 (0%)	8.44 (0%)	2.51 (0%)
$T_{p}$ = 1.7	0.76 (−28%)	8.66 (0%)	8.45 (0%)	2.52 (0%)
$T_{p}$ = 1.8	0.70 (−8%)	8.65 (0%)	8.45 (0%)	2.51 (0%)
$T_{p}$ = 2.0	0.68 (−2%)	8.65 (0%)	8.45 (0%)	2.49 (−1%)
$T_{p}$ = 5.0	0.70 (+2%)	8.65 (0%)	8.45 (0%)	2.47 (−1%)
$T_{p}$ = 10	0.70 (0%)	8.65 (0%)	8.45 (0%)	2.47 (0%)

**Table 5 sensors-23-03711-t005:** rms Values of key system outputs for simulations with added uncertainty in the wind preview information used by MPC.Here Baseline refers to control configuration without wind preview.

Uncertainty	$x_{\mathrm{rms}}$ (cm)	${\left(U_{2}\right)}_{\mathrm{rms}}$ (deg)	$\theta_{\mathrm{rms}}$ (deg)	$q_{\mathrm{rms}}$ (deg s^−1^)
Baseline	14.89	4.20	4.26	2.99
$\Delta_{t}=0.5$ s	4.84	3.98	4.06	2.84
$\Delta_{t}=1.0$ s	9.56	4.09	4.12	3.05
$\Delta_{t}=1.5$ s	13.94	4.18	4.26	3.17
$\Delta_{t}=1.75$ s	15.93	4.22	4.31	3.19
$\Delta_{t}=2.0$ s	17.68	4.25	4.34	3.18
$\Delta_{g}=+50$%	5.73	3.75	3.83	2.44
$\Delta_{g}=+100$%	12.55	3.63	3.68	2.31
$\Delta_{g}=+120$%	15.30	3.59	3.62	2.26
$\Delta_{g}=+150$%	19.35	3.52	3.53	2.21

## Data Availability

Not applicable.

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
