# Peer review of "Wind Preview-Based Model Predictive Control of Multi-Rotor UAVs Using LiDAR"

_sensors, 2023, doi:10.3390/s23073711_

Round 1

Reviewer 1 Report

In this paper, a model predictive controller equipped with wind preview is proposed to control a quad-rotor mainly during take off and landing. Simulation results show that the controller is successful in disturbance rejection. 

The paper is well structured and well written. Literature review addresses almost all important items. Block diagrams, formulations and discussions all sound. A good piece of work to be published indeed. Just one point for more clarification:

1- In the abstract, authors mentioned 80% performance improvement compared to sole feedback based disturbance rejection. As far as I understand from Fig. 3, the comparison is between the diagram with and without MPC block. Am I right? Please clarify what "sole feedback based disturbance rejection" exactly mean.

Reviewer 2 Report

Dear Authors,

congratulations on the work under consideration. The article has merits. Here, I present some remarks about your research.

In the introduction section, there are few references regarding the use of LiDAR in UAVs, even though it is a subject little explored in the literature.

The possibility of replicating the experimental results is relevant for the scientific community, however, the article lacks some important information:

·       What are the measured state variables? (figure 3 has this information but it is not detailed and shown in the text as the states, line 196, and the control inputs, line 205).

·       Which sensors are present in the quadrotor?

·       What is the sampling time used in the discretization.

·       What are the values and how some parameters of the PID controller (proportional, integral and derivative gains) and the MPC (control horizon and weighting matrices) were adjusted?

·       What is the structure and parameters of the selected KF and why a covariance equal to 0.3 of the white noise was used?

The presentation of practical results during the landing and takeoff would extremely improve this article, since it has already been demonstrated, in past research, the possibility of remote sensing of incoming wind gusts. These results would replace some of those presented in the article, especially those associated with the linear model, which do not have a significant scientific relevance in the practical context of the work.

As stated in the introduction section, there are different methods for the attenuation of the wind effects during the aircraft flight, especially those that estimate this disturbance. Therefore, it would be interesting to compare the proposed method, which was mentioned as a contribution of the article, with other recent ones in the literature. This comparison becomes even more relevant because this research uses LiDAR as an exclusive additional equipment for wind disturbances.

Other points:

·       Would the vector wb be transposed? (line 161).

·       Difference between the name of the quadrotor in figure 2 and line 184.

·       Not described what the vector w is in the equation in line 193.

·       The units of the variables in table 1 are missing.

·       Check the text in line 240.

·       It was not specified what Δ represents in line 236.

·       Check equation of the lines 242 and 244.

·       Avoid using the first person as in lines 249 and 250.

Best regards,

Reviewer 3 Report

1. At the end of line 168 on page 5, a full stop is missing.

2. Don't add a comma after "give by" in line 172 on page 5.

3. It can appropriately supplement the comparison between the effect of this model and that of other models.

Reviewer 4 Report

In this study the development of a linear MPC with wind disturbance preview, the analysis of the MPC’s performance against varying preview lengths, the verification of the proposed performance benefits through non-linear simulations using experimental LiDAR data, and a preliminary analysis into the robustness of the system against uncertainty in the wind preview information were developed.

Generally, the text is well written but contains many acronyms that are not easy to read. For this reason, if the journal allows it, I suggest inserting a table at the beginning with the explained acronyms present throughout the text.

In addition, it does not follow a real scientific research scheme: for example, both the context (i.e., precision farming which sees UAVs as protagonists) and the multivariate statistical analysis used (i.e., predictive models) are not well specified.

For all these reasons, I will reconsider the article for the publication only after major revisions of the questions reported below.

Minor comments

Keywords

Do not use the same words as in the title (e.g., UAV).

Major comments

Introduction

As mentioned before, UAV technology and precision agriculture are very close. Please update the entire text with PA references and contextualize it. It could permit a greater credibility to the whole work.

Specify that there are two types of UAVs that can be used: light (without the use of licenses) and not (having several restrictions) and report their differences.

Predictive models (MPC)

- there is not a real introduction on this topic, please report it;

- specify whether statistical or deterministic predictive models are used in this study;

- there are no examples of the use of these provisional models in literature (insert them to better contextualize the study);

- the fact that the predictive model used in this work is in real time is not very marked. Specify this.

Update the references.

Methods and Systems

In the paragraph all the technical specifications of the UAV used are difficult to read. Report, for example, a table like the following (it is only an example):

Details

Items

Specifications

Light drone

Weight

Dimensions

Max speed

Satellite positioning systems

Digital camera

Camera

Sensor Resolution

Image Sensor Type

Capture Formats

Still Image Formats

Video Recorder Resolutions

Frame Rate

Still Image Resolutions

GIMBAL

Control range Inclination

Stabilization

Obstacle detection distance

Operating environment

Remote Control

Operating Frequency

Max Operating Distance

Battery

Supported Battery Configurations

Rechargeable Battery

Technology

Voltage Provided

Capacity

Run Time (Up To)

Recharge Time

Conclusions

This part is really confusing. Try to report a workflow to help the reader better understand the results obtained in this study.

Round 2

Reviewer 2 Report

Dear Authors,

The authors did not send any files answering my questions. Only minor text changes were made, and I don't know if they were recommendations from other authors. Because regarding my suggestions, they only replaced one unit in a table.

In short, no changes were made, and no questions answered.

I mentioned typos and none of that was done. I wondered if they received the text I wrote for the authors.

Under these conditions I cannot change my recommendation.

I am resending below the previews recommendations so they can read.

Start here

congratulations on the work under consideration. The article has merits. Here, I present some remarks about your research.

In the introduction section, there are few references regarding the use of LiDAR in UAVs, even though it is a subject little explored in the literature.

The possibility of replicating the experimental results is relevant for the scientific community, however, the article lacks some important information:

·       What are the measured state variables? (figure 3 has this information but it is not detailed and shown in the text as the states, line 196, and the control inputs, line 205).

·       Which sensors are present in the quadrotor?

·       What is the sampling time used in the discretization.

·       What are the values and how some parameters of the PID controller (proportional, integral and derivative gains) and the MPC (control horizon and weighting matrices) were adjusted?

·       What is the structure and parameters of the selected KF and why a covariance equal to 0.3 of the white noise was used?

The presentation of practical results during the landing and takeoff would extremely improve this article, since it has already been demonstrated, in past research, the possibility of remote sensing of incoming wind gusts. These results would replace some of those presented in the article, especially those associated with the linear model, which do not have a significant scientific relevance in the practical context of the work.

As stated in the introduction section, there are different methods for the attenuation of the wind effects during the aircraft flight, especially those that estimate this disturbance. Therefore, it would be interesting to compare the proposed method, which was mentioned as a contribution of the article, with other recent ones in the literature. This comparison becomes even more relevant because this research uses LiDAR as an exclusive additional equipment for wind disturbances.

Other points:

·       Would the vector wb be transposed? (line 161).

·       Difference between the name of the quadrotor in figure 2 and line 184.

·       Not described what the vector w is in the equation in line 193.

·       The units of the variables in table 1 are missing.

·       Check the text in line 240.

·       It was not specified what Δ represents in line 236.

·       Check equation of the lines 242 and 244.

·       Avoid using the first person as in lines 249 and 250.

Best regards,”

 End here

Best regards,

Reviewer 4 Report

The manuscript in the present form can be published in Sensors as the authors have been significantly improved it. Parts that better contextualize the work have been added (e.g., precision agriculture). The methodology has been better clarified with tables and all is now clearer.

I don’t need to review another version because I accept the work in the present form.

Author Response

.

Round 3

Reviewer 2 Report

Dear Authors,

congratulations on the work under consideration again.

I have received your answers and I agree with all the changes made. The practical results and comparisons with other control methods would greatly improve your work. I hope that in the future these results can be published, as they are relevant to the topic addressed in the research.

Best regards,

Author Response

.